# Turbulence Dissipation Rate Estimated from Lidar Observations During the LAPSE-RATE Field Campaign

Miguel Sanchez Gomez[1], Julie K. Lundquist[1,2], Petra M. Klein[3,4], and Tyler M. Bell[3,4]

[1]Department of Atmospheric and Oceanic Sciences, University of Colorado Boulder, Boulder, 80303, United States
[2]National Renewable Energy Laboratory, Golden, 80401, United States
[3]University of Oklahoma School of Meteorology, Norman, 73072, United States
[4]University of Oklahoma Center for Autonomous Sensing and Sampling, Norman, 73072, United States

*Correspondence to*: Miguel Sanchez Gomez (misa5952@colorado.edu)

**Abstract.** The International Society for Atmospheric Research using Remotely-piloted Aircraft (ISARRA) hosted a flight week in July 2018 to demonstrate Unmanned Aircraft Systems' (UAS) capabilities in sampling the atmospheric boundary layer. This week-long experiment was called the Lower Atmospheric Profiling Studies at Elevation – a Remotely-piloted Aircraft Team Experiment (LAPSE-RATE) field campaign. Numerous remotely piloted aircrafts and ground-based instruments were deployed with the objective of capturing meso- and microscale phenomena in the atmospheric boundary

layer. The University of Oklahoma deployed one Halo Streamline lidar and the University of Colorado Boulder deployed two Windcube lidars. In this paper, we use data collected from these Doppler lidars to estimate turbulence dissipation rate throughout the campaign. We observe large temporal variability of turbulence dissipation close to the surface with the Windcube lidars that is not detected by the Halo Streamline. However, the Halo lidar enables estimating dissipation rate within the whole boundary layer, where a diurnal variability emerges. We also find a higher correspondence in turbulence dissipation

between the Windcube lidars, which are not co-located, compared to the Halo and Windcube lidar that are co-located, suggesting a significant influence of measurement volume on the retrieved values of dissipation rate. This dataset have been submitted to Zenodo (Sanchez Gomez and Lundquist, 2020) for free and open access (DOI:10.5281/zenodo.4399967).

## 1 Introduction

The Lower Atmospheric Profiling Studies at Elevation – a Remotely-piloted Aircraft Team Experiment (LAPSE-RATE) field

campaign took place in July 2018 in the San Luis Valley of Colorado in Western North America (de Boer et al., 2020b). This project aimed to sample the lower atmospheric boundary layer using Unmanned Aircraft Systems (UAS) and ground-based sensors within complex terrain (de Boer et al., 2020a). Each day of the field campaign had specific science themes. The morning boundary layer transition, aerosol properties and their variability, valley drainage flows, deep convection initiation, and turbulence profiling were the major scientific topics sampled throughout the project.

Turbulence measurements in the atmospheric boundary layer are scarce and are usually constrained to the surface layer even as turbulence metrics are critical for applications like wind energy (Bodini et al., 2019b; Lundquist and Bariteau, 2015;

Wildmann et al., 2019), aviation (Muñoz-Esparza et al., 2018; Sharman and Pearson, 2017), and atmospheric aerosol transport (Fernando et al., 2010; Lundquist and Chan, 2007). Furthermore, turbulence measurements are used in improving low-level turbulence forecasting in numerical weather prediction models. UAS and remote sensors offer high resolution measurements

35 of atmospheric variables that can be used to expand turbulence measurements across multiple vertical and horizontal scales. One turbulence proxy that has been linked to high variability of hub-height wind speed predicted by the Weather and Research Forecasting (WRF) model is dissipation of turbulence kinetic energy (Berg et al., 2019; Yang et al., 2017). Turbulence dissipation rate displays high temporal and spatial variability in complex terrain (Bodini et al., 2019a). The dissipation of turbulence kinetic energy is usually evaluated using in-situ high-temporal resolution measurements (Lundquist and Bariteau,

40 2015; Oncley et al., 1996; Piper and Lundquist, 2004). However, recent studies have shown remote sensors can also be used to derive turbulence metrics (Bodini et al., 2018, 2019a; O'Connor et al., 2010; Wildmann et al., 2019).

Three Doppler lidars were deployed during the LAPSE-RATE field campaign to sample the lower atmospheric boundary layer (Bell et al., 2020). Here, we retrieve turbulence dissipation rate from these lidar measurements. We provide a brief description of the platforms deployed and used herein in Section 2. Section 3 describes the procedure for estimating turbulence dissipation

45 rate from lidar measurements, and we show some sample data on Section 4. In Section 5 we perform an uncertainty analysis in the estimation of turbulence dissipation rate. Finally, Sections 6 and 7 are dedicated to a summary and data availability, respectively.

## 2 Observations

Three Doppler lidars were deployed in the San Luis Valley to provide a reference dataset for UAS observations. The University

50 of Oklahoma (OU) deployed the Collaborative Lower Atmospheric Mobile Profiling System (CLAMPS) that contains a Halo Streamline scanning Doppler Lidar, a HATPRO microwave radiometer (Rose et al., 2005), and an Atmospheric Emitted Radiance Interferometer (Wagner et al., 2019). The University of Colorado Boulder (CU) deployed two Leosphere/NRG Version 1 Windcube profiling Doppler lidars (Aitken et al., 2012; Rhodes and Lundquist, 2013). Table 1 presents the technical specifications for the lidars.

55 **Table 1. Main technical specifications of the lidars used in this study.**

|  | Windcube v1 (WC49 & WC68) | Halo Streamline |
|---|---|---|
| Wavelength $[\mu m]$ | 1.54 | 1.54 |
| Receiver bandwidth $[MHz]$ | $\pm\,55$ | $\pm\,25$ |
| Nyquist velocity $[m\ s^{-1}]$ | 42.3 | 19.4 |
| Signal spectral width $(\Delta v)\ [m\ s^{-1}]$ | 3.39 | 1.5 |
| Pulses averaged $(n)$ | 10 000 | 20 000 |
| Points per range gate $(M)$ | 25 | 10 |

| | | |
|---|---|---|
| Vertical resolution [$m$] | 20 | 20 |
| Minimum range gate [$m$] | 20 | 15 |
| Number of range gates | 10 | 320 |
| Pulse width [$ns$] | 200 | 150 |
| Time resolution [$Hz$] | ~1 | 1 |

The Halo Streamline and one Windcube v1 (hereafter referred to as WC49) lidars were generally located at the Moffat site. The remaining Windcube lidar from CU (hereafter referred to as WC68) was located at the Saguache site, although the CU lidars were collocated for a brief period from 14 July 00:04 UTC to 21:44 UTC at Saguache. The relative location of each site
60   is shown in Figure 1.

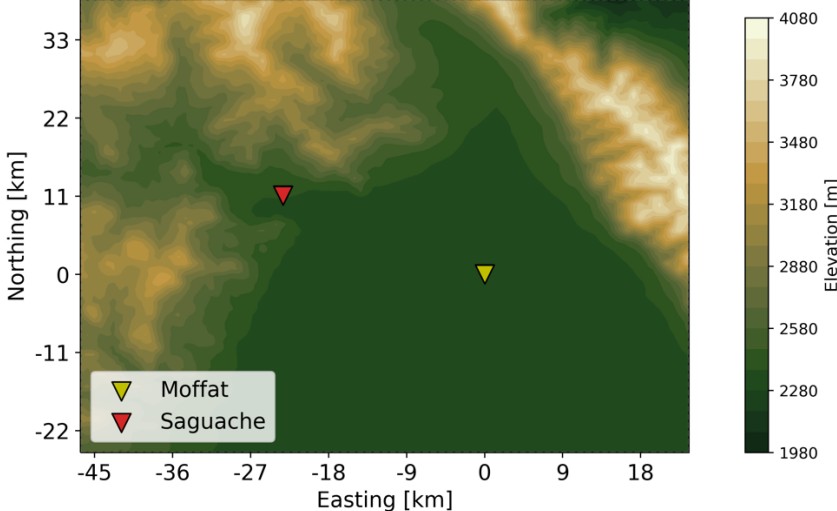

**Figure 1. Map of the relevant locations during the LAPSE-RATE field campaign. Elevation contours are shown every 100 m. Distances measured from the Moffat site.**

The scan strategy for the Halo Streamline consisted of a 24-point plan position indicator (PPI) scan at 70 degree elevation, a
65   6-point PPI at 45 degrees, and a vertical stare. Here, we use the 24-point scan to estimate the horizontal wind vector at a 5-minute temporal resolution, and the vertical stare to derive turbulence characteristics of the flow at a ~1.5-second temporal resolution. The Windcube lidars measure line-of-sight velocity along the four cardinal directions at a 62 degree elevation. The wind vector is estimated every 4 seconds using the Doppler Beam Swinging approach (Lundquist et al., 2015). A more complete description of the three platforms can be found in Bell et al. (2020) including quality control on the data.

## 3 Turbulence Dissipation Rate from Doppler Lidars

Doppler lidars offer insight into the variability of turbulence dissipation rate in complex terrain. Moreover, lidars provide atmospheric measurements at higher altitudes than in-situ sensors, extending the spatial sampling of the boundary layer beyond surface-based towers.

We estimate turbulence dissipation rate ($\varepsilon$) from the variance of the line-of-sight velocity following the methodology proposed by O'Connor et al. (2010) and refined by Bodini et al. (2018). The turbulence energy spectrum within the inertial subrange for homogeneous and isotropic turbulence can be expressed as a function of wavenumber $k$ (Kolmogorov, 1941):

$$S(k) = a\varepsilon^{2/3}k^{-5/3},\tag{1}$$

where $a \approx 0.52$ is the one-dimensional Kolmogorov constant. Eq. (1) can be integrated over the wavenumber space within the inertial subrange. Furthermore, Taylor's frozen turbulence hypothesis relates the wavenumber $k$ with a lengthscale $L = 2\pi/k$, resuling in the following:

$$\sigma_v^2 = \int_k^{k_1} S(k)dk = -\frac{3}{2}a\varepsilon^{\frac{2}{3}}\left(k_1^{-\frac{2}{3}} - k^{-\frac{2}{3}}\right) = \frac{3}{2}a\left(\frac{\varepsilon}{2\pi}\right)^{\frac{2}{3}}\left(L_N^{\frac{2}{3}} - L_1^{\frac{2}{3}}\right),\tag{2}$$

where $L_1$ is the length scale for a single lidar sample, $L_N$ is the length scale for a number $N$ of samples used for the calculation ($L_N = NL_1$), and $\sigma_v^2$ is the variance of the de-trended line-of-sight velocity from $N$ samples. The length scale $L_1$ is defined as

$$L_1 = Ut + 2z\sin\left(\frac{\theta}{2}\right),\tag{3}$$

where $U$ is the horizontal wind speed, $t$ is the dwell time, $z$ is the height above the surface, and $\theta$ is the half-angle divergence of the lidar beam. Doppler lidars generally have $\theta < 0.1$ mrad, and so the second term in Eq. (3) can be neglected.

This method for estimating the rate of dissipation of turbulence kinetic energy is based on the assumption that the samples used in the calculation reside within the inertial subrange of turbulence. Therefore, if either a too long or too short sample length $N$ is used, an underestimation or overestimation of $\varepsilon$, respectively, will occur (Bodini et al., 2018). We estimate the sample length by determining the extension of the inertial subrange from an experimental fit to the turbulence spectrum.

### 3.1 Windcube v1 Lidar

For the Windcube lidars, $\sigma_v^2$ is estimated as the average from the four beams and horizontal wind speed $U$ is derived from the line-of-sight measurements from the different beams.

The Windcube lidars operated in profiling mode rather than in vertical stare mode, we therefore estimate the sample length $N$ from the 15-minutes turbulence spectrum using a power fit to the data (Figure 2). We filter out frequencies greater than 0.05 Hz, which are affected by instrumental noise (Frehlich, 2001). We find the power fit to the data between each frequency ($f < 0.05\ Hz$) and the cut-off frequency (0.05 Hz). Then, the sample length is obtained from the frequency that gives a power fit with the best agreement with Kolmogorov $f^{-5/3}$ law. In such a way, we force the sample length to be calculated from the portion of the spectrum that shows the highest agreement with the energy cascade. A statistical analysis on the difference

100     between the power law fit for each spectrum and the Kolmogorov law shows only the upper range gates diverge from the expected $f^{-5/3}$ law for both nighttime and daytime conditions (not shown). Figure 2 shows the turbulence spectrum estimated from the Windcube lidars when they were co-located at the Saguache site. The WC49 and WC68 lidars exhibit the transition frequency at $3.3 \times 10^{-3}\ s^{-1}$ and $1.1 \times 10^{-2}\ s^{-1}$, respectively, for daytime conditions (Figure 2a,b), and $7.7 \times 10^{-3}\ s^{-1}$ and $2.65 \times 10^{-2}\ s^{-1}$, respectively, for nighttime conditions (Figure 2c,d).

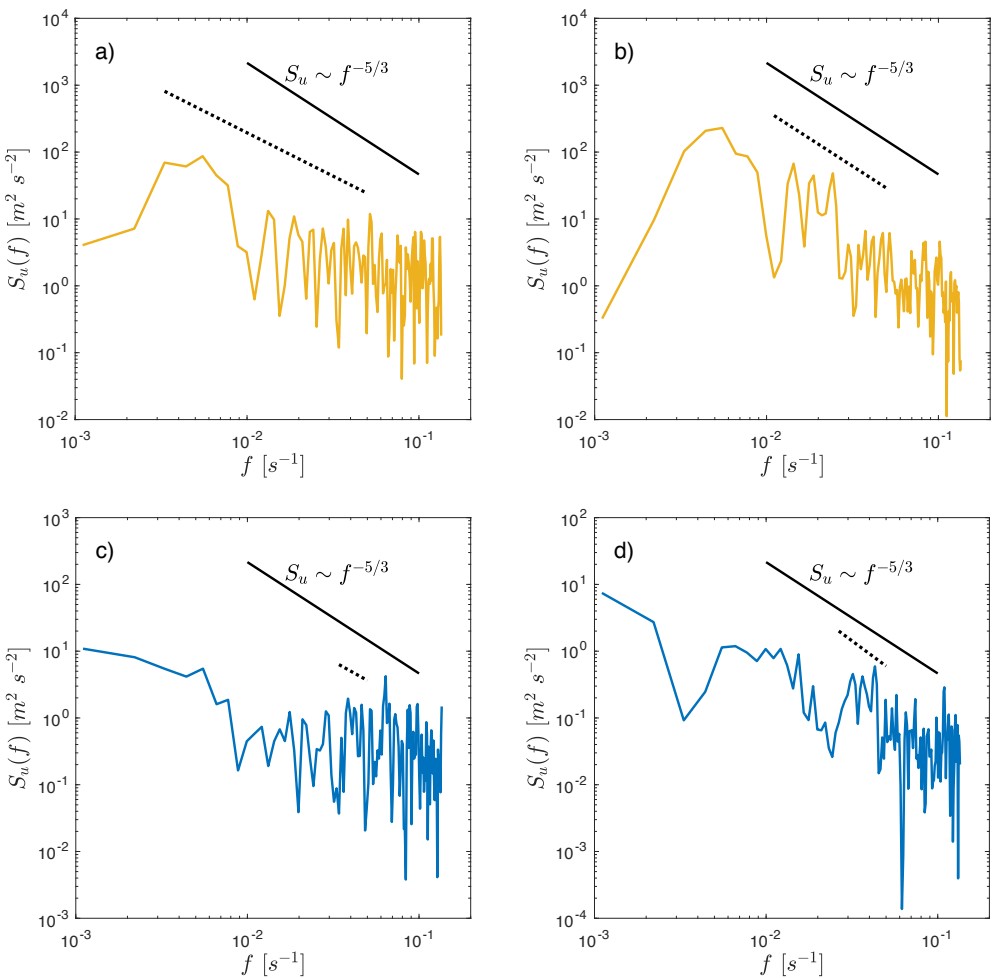

105

**Figure 2. Example of smoothing spline fit of an experimental spectrum of the line-of-sight velocity measured by one beam of the WC49 (a,c) and WC68 (b,d) lidars at 120 m above the surface. The top panels represent daytime conditions on 14 July 17:30 UTC, and the bottom panels represent nighttime conditions on 14 July 07:00 UTC. Turbulence dissipation at this time was**
110     **$6.43 \times 10^{-3}\ m^2\ s^{-3}$ for the WC49, and $6.15 \times 10^{-3}\ m^2\ s^{-3}$ for the WC68. The solid black line shows the theoretical -5/3 slope of the spectrum in the inertial subrange. The dotted black line shows the slope of the power fit to the spectrum, starting from the transition frequency and ending at 0.05 Hz .**

As expected, daytime conditions evidence smaller transition frequencies to the inertial subrange compared to nighttime conditions for both lidars (Figure 3). During the day, convective conditions contribute energy at larger scales than during the

night. Transition frequencies to the inertial subrange are consistently smaller for the WC68 lidar compared to the WC49 lidar, especially during the night. The WC68 lidar is closely surrounded by complex terrain and the exit of a valley that modifies turbulence close to the surface, whereas the WC49 lidar is located further away from terrain features.

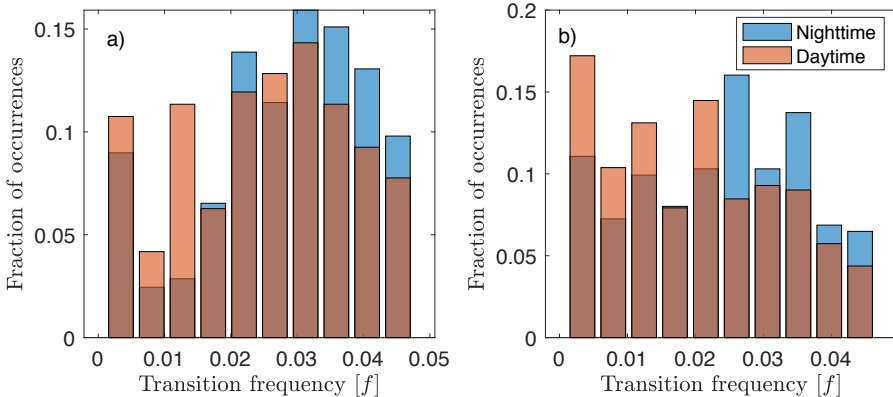

**Figure 3. Transition frequency distributions for the WC49 (a) and WC68 (b) lidars during nighttime and daytime conditions.**

### 3.2 Halo Streamline Lidar

For the Halo Streamline, the variance of the line-of-sight velocity $\sigma_v^2$ is calculated from the vertically pointing beam and wind speed $U$ is retrieved from a sine-wave fit to the vertical-azimuth display (VAD) scans every 5 minutes as provided by Bell et al. (2020).

We estimate the sample length $N$ by fitting the general kinematic spectral model for a vertical velocity field proposed by Kristensen et al. (1989) to the vertical velocity spectra:

$$\frac{S(k)}{\sigma_w^2} = \frac{l_w}{2\pi} \frac{1 + \frac{8}{3}\left(\frac{l_w k}{a(\mu)}\right)^{2\mu}}{\left[1 + \left(\frac{l_w k}{a(\mu)}\right)^{2\mu}\right]^{\frac{5}{6\mu}+1}}, \tag{4}$$

where

$$a(\mu) = \frac{\pi\mu\Gamma\left(\frac{5}{6\mu}\right)}{\Gamma\left(\frac{1}{2\mu}\right)\Gamma\left(\frac{1}{3\mu}\right)}, \tag{5}$$

and $\mu$ regulates the curvature of the turbulence spectrum, we use $\mu = 1.5$ as recommended by Lothon et al. (2009). The turbulence spectrum $S(k)$ is obtained from 15 minutes of data and we filter out frequencies greater than 0.08 Hz, which are affected by instrumental noise (Frehlich, 2001). The transition wavelength to the inertial subrange can be written as a function of the integral scale $l_w$ and $\mu$ as follows:

$$\lambda_w = \frac{2\pi l_w}{a(\mu)} \left\{ \frac{5}{3} \sqrt{\mu^2 + \frac{6}{5}\mu + 1} - \left( \frac{5}{3}\mu + 1 \right) \right\}^{1/2\mu}, \tag{6}$$

The sample length is obtained by dividing the transition wavelength by the wind speed derived from the closest VAD scan and
vertical velocity dwell time. Figure 4 shows an example of the experimental and modelled turbulence spectrum for the Halo lidar.

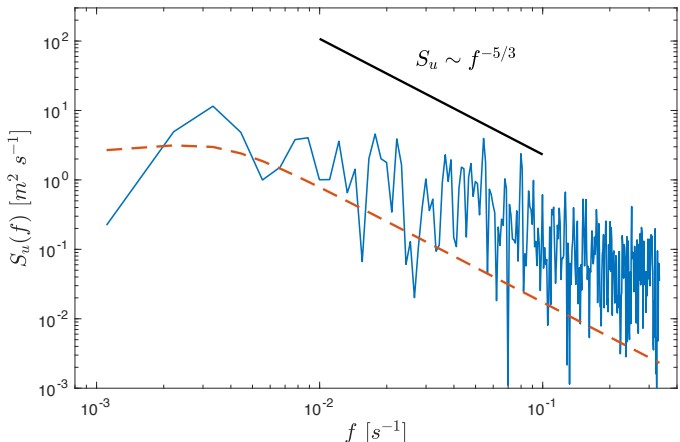

**Figure 4. Example of model fit to experimental spectrum of the line-of-sight velocity measured by the Halo Streamline lidar at 105 m above the surface on 15 July 01:23 UTC. Turbulence dissipation at this time was $1.18 \times 10^{-2} \ m^2 \ s^{-3}$. The red dashed line shows**
**the spectrum fit. The solid black line shows the theoretical -5/3 slope of the spectrum in the inertial subrange.**

### 3.2 Noise Removal

The final step in estimating $\varepsilon$ is removing the contribution of instrumental noise to the velocity variance. The velocity variance calculated from lidar measurements is a combination of atmospheric turbulence $\sigma_U^2$, instrumental noise $\sigma_e^2$, and variations in the aerosol terminal fall velocity in the sampled volume $\sigma_d^2$, all of which can be assumed to be independent of one another:

$$\sigma_v^2 = \sigma_U^2 + \sigma_e^2 + \sigma_d^2, \tag{7}$$

According to Pearson et al. (2009), the contribution from instrumental noise can be expressed as follows:

$$\sigma_e^2 = \frac{\Delta v^2 \sqrt{8}}{\alpha N_p} \left( 1 + \frac{\alpha}{\sqrt{2\pi}} \right)^2, \tag{8}$$

where the lidar photon count to speckle count $(\alpha)$ can be estimated as a function of bandwidth $(B)$:

$$\alpha = \frac{SNR}{\sqrt{2\pi}} \frac{B}{\Delta v}, \tag{9}$$

and the accumulated photon count $(N_p)$ can be estimated as a function of the lidar Signal-to-Noise-Ratio $(SNR)$ as:

$$N_p = SNRnM. \tag{10}$$

See Table 1 for a complete description of the nomenclature.

Therefore, turbulence dissipation rate can be calculated, using the appropriate sample length $N$, by means of Eq. (11).

$$\varepsilon = 2\pi \left(\frac{2}{3a}\right)^{\frac{3}{2}} \left(\frac{\sigma_v^2 - \sigma_e^2}{L_N^{\frac{2}{3}} - L_1^{\frac{2}{3}}}\right)^{3/2} . \tag{11}$$

## 4 Sample Data

Turbulence dissipation rate estimated from each lidar exhibits large variability throughout the whole field campaign (Figure 5). High values of $\varepsilon$ ($\sim 10^{-3}$) generally occur during the day and low values ($\sim 10^{-4}$) occur during nighttime, as seen by lidar estimates in other locations (Bodini et al., 2018, 2019a; Muñoz-Esparza et al., 2018; Wildmann et al., 2019). Furthermore, the surface layer exhibits larger dissipation than the boundary layer (Figure 5a), consistent with the in-situ measurements of Balsley et al. (2006). Data for the first three range gates in the Halo lidar were discarded because the vertical velocity variance remained constant in time. This artifact is common on Halo Doppler Lidars, and has been previously reported (Pearson et al., 2009).

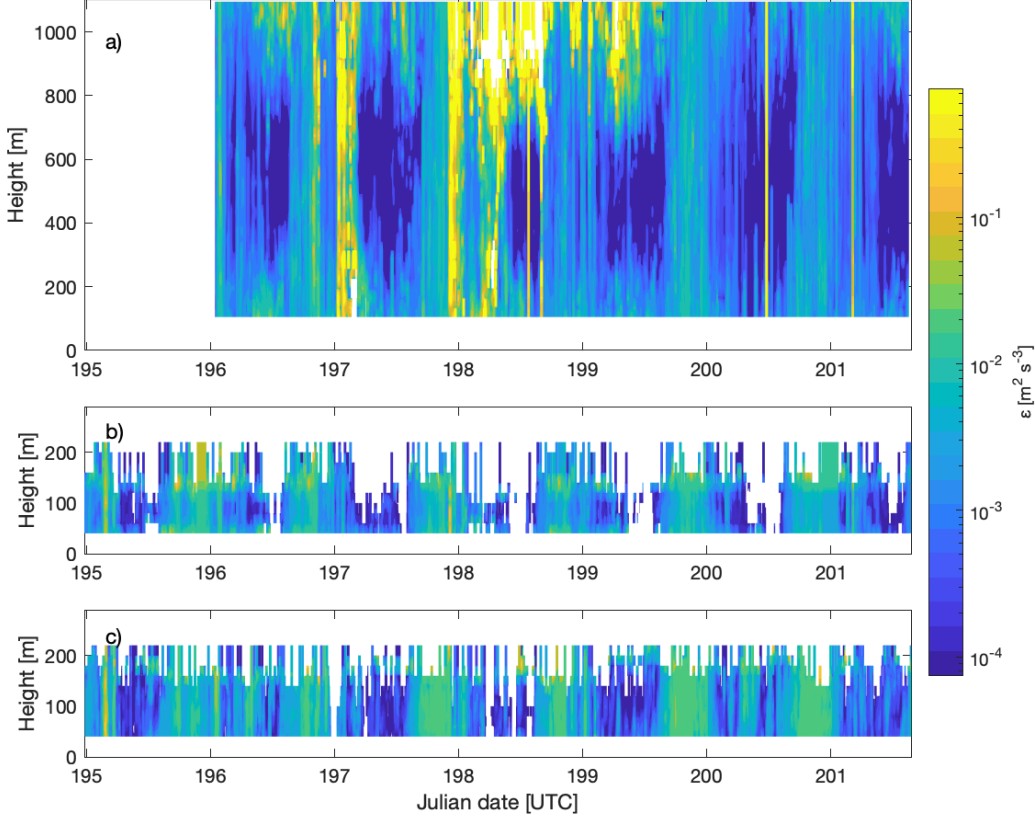

Figure 5. Time-height contours of 30-min averaged turbulence dissipation rate for the Halo Streamline (a), Windcube WC49 (b), and Windcube WC68 (c) for the duration of the field campaign.

The Windcube lidar located at the Moffat site (WC49) suffers from some data gaps in the estimation of turbulence dissipation (Figure 5b). This lidar exhibits low Carrier-to-Noise-Ratio (CNR) values throughout the sampling period, especially during nighttime (Figure 6). Therefore, the contribution of instrumental noise to the velocity variance gives in $\sigma_e^2 > \sigma_v^2$, invalidating the assumptions made in Eq. (7).

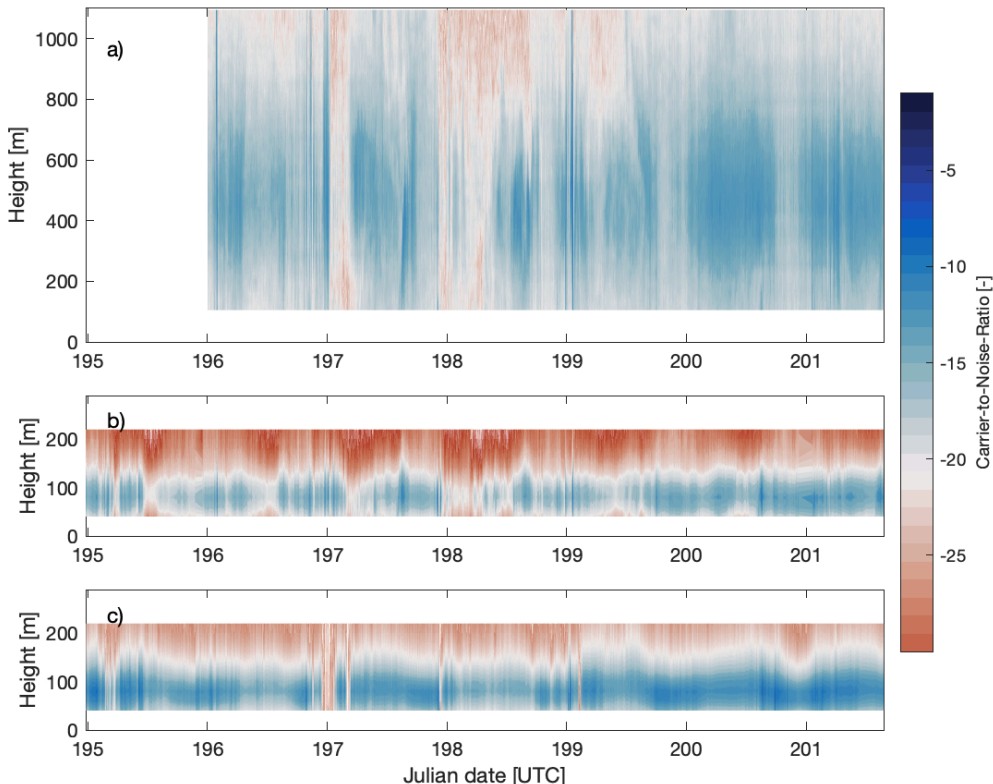


**Figure 6. Time-height contours of Carrier-to-Noise-Ratio for the vertical velocity measured by the Halo Streamline (a), the horizontal velocity measured by the Windcube WC49 (b), and the horizontal velocity measured by the Windcube WC68 (c) for the duration of the field campaign. Data with CNR values above -21 are used for the calculations (blue shading), whereas data with CNR below -21 are discarded (red shading).**

The two lidars co-located at the Moffat site provide the opportunity to compare turbulence dissipation rates calculated from different platforms (Figure 7), as was previously done with the Halo and Windcube lidars (Bodini et al., 2018) at a different location with different scanning strategy for the Halo. The Pearson correlation coefficient for $\log_{10} \varepsilon$ recorded at 105 m, 135 m, and 160 m between both lidars at Moffat is 0.26851, 0.19065, and 0.12751, respectively. Furthermore, the $R^2$ values for these heights are 0.0721, 0.0363, and 0.0163, respectively. This poor agreement demonstrates that the assumptions and

methodology for estimating $\varepsilon$ from two scanning techniques provide very different results within the surface layer. One source of discrepancy in the calculations of $\varepsilon$ is the horizontal velocity estimated from the VAD scan for the Halo Streamline and from the four beams for the Windcube lidars. The horizontal velocity for the VAD scan is estimated every 5 minutes, removing high wind speed gusts, whereas horizontal wind speed is calculated every 4 seconds for the Windcubes. Also, dissipation rate

estimated from the Halo Streamline displays much larger values and higher variability over shorter time scales, whereas the
Windcube evidences larger variability over longer time periods.

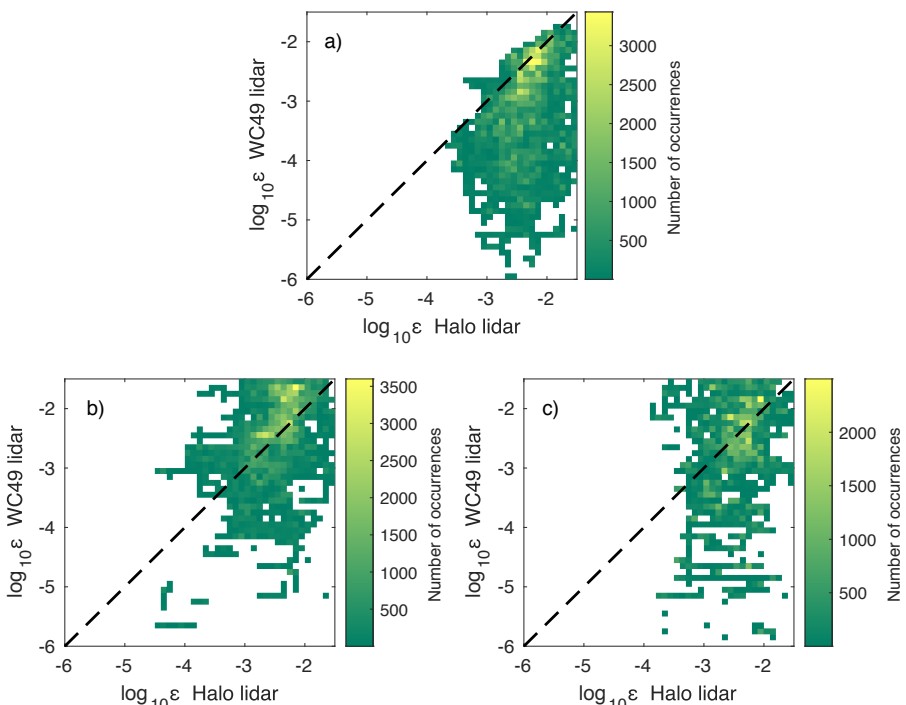

**Figure 7. Heat map of $\log_{10}\varepsilon$ showing scanning technique variability at the Moffat site for both lidars at 105 m (a), 135 m (b), and 160 m (c). The shading indicates the number of occurrences of each value of $\log_{10}\varepsilon$ for each lidar. The values of turbulence dissipation rate are averaged over a 30-minute time period. The black dashed line shows a 1:1 relationship.**

Turbulence dissipation rate also varies spatially (Figure 8). Surprisingly, the Pearson correlation coefficient between the two
Windcube lidars at the different sites is 0.64637, 0.6388, 0.43664, 0.37147 at 75 m, 105 m, 135 m, and 160 m, respectively.
Furthermore, the $R^2$ values for these heights 0.4178, 0.4081, 0.1907, 0.1380, respectively. The higher correlation between
both Windcube lidars compared to the lidars at the Moffat site may be partly explained by their same scanning technique (and
resulting temporal averaging) and dissimilar weighting function between the Halo and Windcube lidars. Also, both Windcube
lidars display a strong diurnal variability in turbulence dissipation close to the surface that is not observed in the Halo lidar,
perhaps due to the Halo's larger measurement volume (Figure 5).

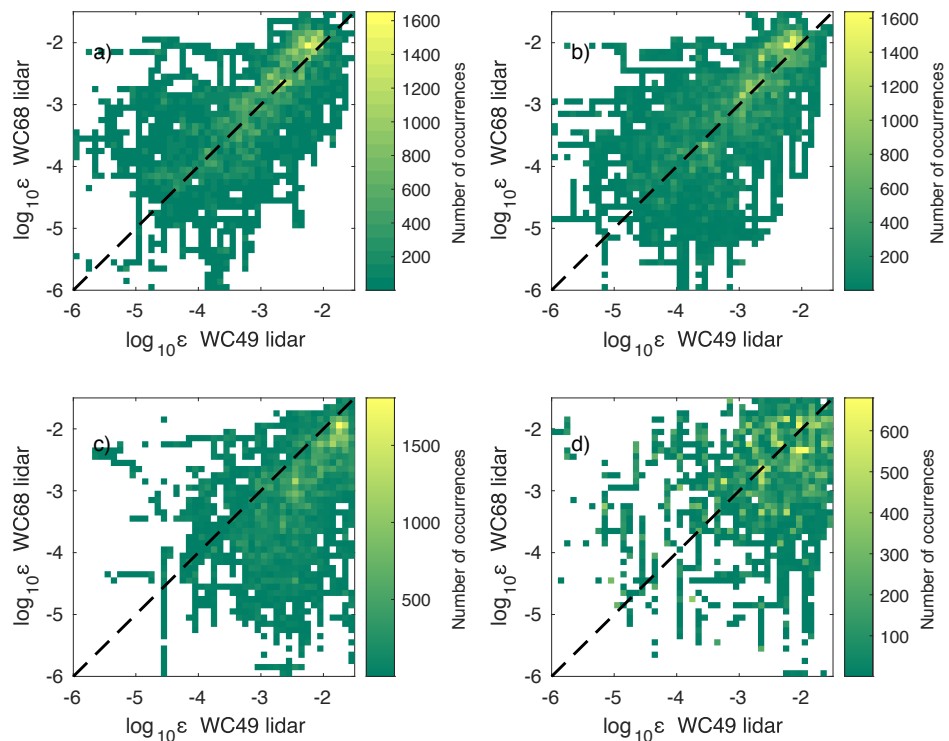

**Figure 8.** Heat map of $\log_{10} \varepsilon$ showing the spatial variability for both Windcube lidars at 75 m (a), 105 m (b), 135 m (c), and 160 m (d). The shading indicates the number of occurrences of each value of $\log_{10} \varepsilon$ for each lidar. The values of turbulence dissipation rate are averaged over a 30-minute time period. The black dashed line shows a 1:1 relationship.

We observe a higher correlation in turbulence dissipation for all lidars during the day compared to the night (Figure 9). Although there is a higher correlation between both Windcube lidars during daytime ($R^2 = 0.56$ for daytime and $R^2 = 0.02$ for nighttime), the Pearson correlation coefficient suggests $\varepsilon$ follows the same trend during the day and night ($R = 0.75$ for daytime and $R = 0.14$ for nighttime). For both lidars located at the Moffat site, we obtain a Pearson correlation coefficient of 0.25 during daytime and 0.32 during nighttime for $\varepsilon$ estimated at 100 m above the ground. Moreover, a straight line fit to the data suggests $\varepsilon$ estimated from the Halo lidar is consistently several orders of magnitude larger than $\varepsilon$ estimated from the Windcube.

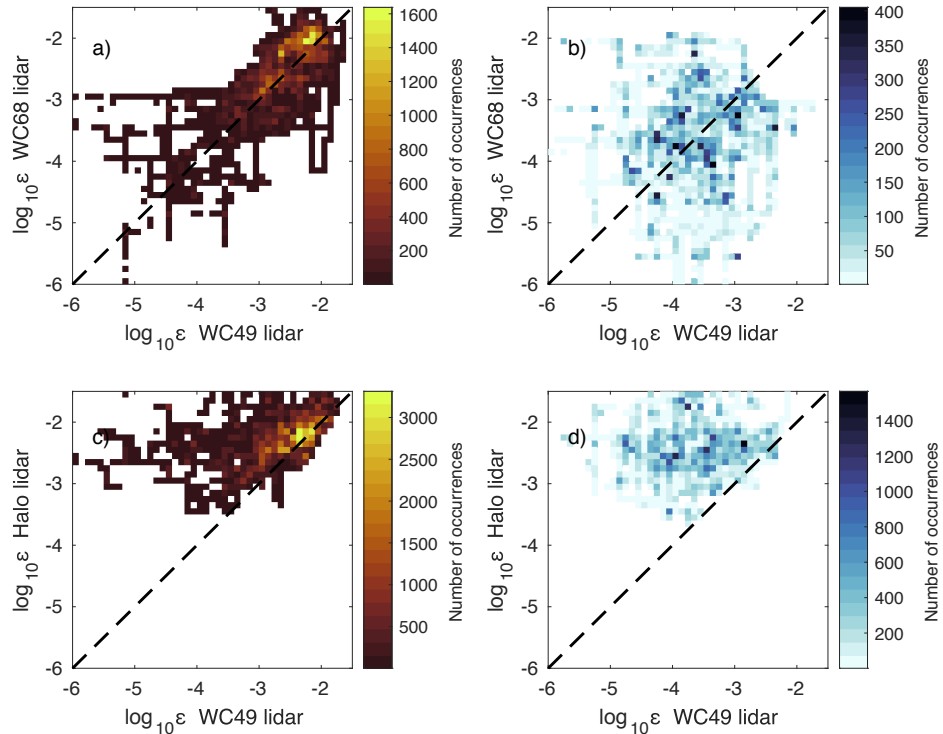

**Figure 9. Heat map of $\log_{10} \varepsilon$ estimated at 100 m showing the spatial variability for both Windcube lidars (a, b), and the platform variability for the Windcube and Halo lidars (c, d). The shading indicates the number of occurrences of each value of $\log_{10} \varepsilon$ for each lidar, warm shading corresponds to daytime and cool shading to nighttime. The values of turbulence dissipation rate are averaged over a 30-minute time period. The black dashed line shows a 1:1 relationship.**

## 5 Uncertainty Analysis

We apply the law of combination of errors to estimate the uncertainty in the retrievals of $\varepsilon$ from random error propagating through our calculations (Barlow, 1989). Assuming $f(x_i)$ is a function of $x_i$ independent and uncorrelated variables, then the variance in $f$, approximated by $\sigma_f^2$ is given by

$$\sigma_f^2 = \left(\frac{\partial f}{\partial x_i}\right)^2 \sigma_{x_i}^2, \tag{12}$$

where $\sigma_{x_i}^2$ are the sample variances in the $x_i$ variables. Using this method, we estimate the uncertainty in the retrievals of $\varepsilon$ from the uncertainty in the line-of-sight velocity variance:

$$\sigma_{\varepsilon,v} = \left(\frac{\partial \varepsilon}{\partial \sigma_v}\right)^2 \sigma_{\sigma,v}^2$$

$$= 2\pi \left(\frac{2}{3\alpha}\right)^{\frac{3}{2}} \left(\frac{\sigma_v^2 - \sigma_e^2}{L_N^{\frac{2}{3}} - L_1^{\frac{2}{3}}}\right)^{\frac{1}{2}} \frac{3\sigma_v}{L_N^{\frac{2}{3}} - L_1^{\frac{2}{3}}} \sigma_{\sigma,v}$$

$$= \varepsilon \frac{3\sigma_v}{L_N^{\frac{2}{3}} - L_1^{\frac{2}{3}}} \sigma_{\sigma,v}, \tag{13}$$

where $\sigma_{\sigma,v}$ is the uncertainty in the sample variance. Although $\sigma_{\sigma,v}$ is not directly measured, it is conventionally considered to

225   be of the same order of magnitude as the instrument noise (i.e. $\sigma_e$).

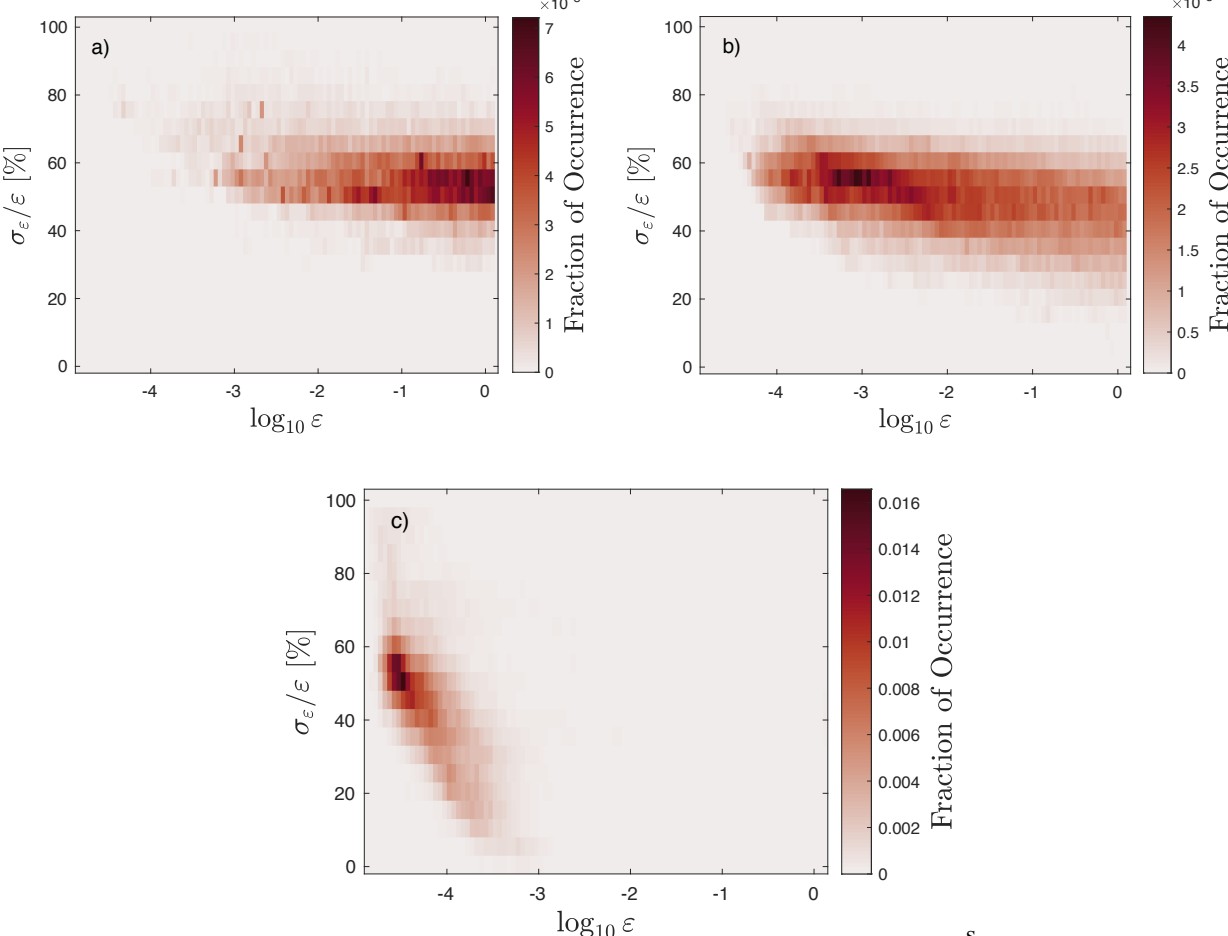

**Figure 10. Fraction of occurrence of each uncertainty estimation and turbulence dissipation rate combination for the WC49 (a), WC68 (b), and Halo (c) lidars.**

230   In general, small values of $\varepsilon$ are associated with large uncertainties (Figure 10). The WC68 and Halo lidars display decreasing uncertainties as the estimated turbulence dissipation rate increases. However, the uncertainty in $\varepsilon$ for the WC68 lidar consistently remains much larger compared to the Halo lidar for all values of $\varepsilon$. Conversely, uncertainties for the WC49 remain

nearly constant for the estimated $\varepsilon$ values. The WC49 lidar exhibits the smallest Carrier-to-Noise ratios for the three lidars throughout the measuring period (Figure 6), which is associated to the instrument noise as shown in Eq. 8.

## 6 Summary

Scientists from multiple research institutions gathered on July 2018 in the San Luis Valley, CO to gather observational data of the atmospheric boundary layer using ground based and aerial platforms. The University of Oklahoma and University of Colorado Boulder deployed three Doppler lidars throughout the campaign to collect vertical profiles of the kinematic state of the atmosphere. Here, we describe the methodology for estimating turbulence dissipation rate using a vertical staring lidar and two profiling lidars, and provide some analysis of the temporal and spatial variability of this metric. We find high temporal variability in turbulence dissipation throughout the whole field campaign. Furthermore, turbulence dissipation rate close to the surface estimated from different platforms at the same site displays significant differences for daytime and nighttime periods. The Halo lidar shows lower variability of turbulence dissipation in the surface layer compared to the Windcube lidar. In contrast, the Windcube lidars, although at several kilometers from one another, exhibit a higher correlation in turbulence dissipation, especially during the day.

## 7 Data Availability

The data in this paper are available for download at https://doi.org/10.5281/zenodo.4399967 (Sanchez Gomez and Lundquist, 2020). The dataset is structured following guidance of de Boer et al. (2020a).

## Author Contributions

MSG performed the calculations, data analysis, and prepared the initial draft. JKL guided the data collection, data processing, and quality control for the Windcube lidars. MSG and JKL contributed to manuscript revision. TB and PK guided the data collection, data processing, and quality control for the Halo Streamline lidar.

## Competing Interests

The authors declare no competing interests.

## Acknowledgements

The authors acknowledge NSF AGS-1554055 (Career) for support for JKL. This work was authored [in part] by the National Renewable Energy Laboratory, operated by Alliance for Sustainable Energy, LLC, for the U.S. Department of Energy (DOE)

under Contract No. DE-AC36-08GO28308. Funding provided by the U.S. Department of Energy Office of Energy Efficiency and Renewable Energy Wind Energy Technologies Office. The views expressed in the article do not necessarily represent the views of the DOE or the U.S. Government. The U.S. Government retains and the publisher, by accepting the article for publication, acknowledges that the U.S. Government retains a nonexclusive, paid-up, irrevocable, worldwide license to publish or reproduce the published form of this work, or allow others to do so, for U.S. Government purposes.

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
