# Peer review of "Turbulence Dissipation Rate Estimated from Lidar Observations During the LAPSE-RATE Field Campaign"

_Earth System Science Data, 2020_

## Author Comment (AC1)

**Response to reviewer 1**

Dear Anonymous Reviewer 1,

we highly appreciate your feedback. It helped us to improve the manuscript. Below we comment on your suggestions in detail.

All reviewer comments appear in *italic text* below, while authors' responses appear in blue text. Line numbers referenced in the authors' responses refer to the revised document.

*The manuscript presents a data set on the turbulent dissipation rate derived from 3 lidar systems during the LAPSE-RATE field campaign in the San Luis Valley, Colorado. Although the calculations of the dissipation rate are based on previously published methods and algorithms is the data set of interest, as it includes the possibility of comparison and validation of the algorithms i) at two different locations (by two identical Leosphere WindCube lidars) and ii) for two different lidar systems at one location (Leosphere Windcube v1 vs. Halo Streamline).*

**General comments**

*The manuscript is in general clearly written and well structured and the data are well described and presented. I see, however one main issue in the lowest layers of the Halo instrument. The yellowish and very constant (at least in height) "bright band" around 50 m looks rather suspicious, and I am rather in doubt that this is an expression of the surface layer as the authors state. I hypothesize that this is some kind of measurement artefact close to the ground. If it would be a real (and of course expected) enhancement due to the surface layer, I would expect a clear diurnal variation in its vertical extension (which I can only see in very weak nuances) and an additional clear dependency on the wind speed. This has to be closer investigated and discussed before I can recommend the manuscript to be considered for publication. I firmly believe there is a measurement/evaluation issue in the lowest range gates for the Halo system. A first important test would be to look into (and also present) two additional time height plots of horizontal and vertical velocity in Fig. 4.*

Thank you for this comment, we looked into this issue and compared our lidar measurements with similar datasets. We found this artifact is common on Halo Doppler Lidars, and also has been previously reported (Pearson et al., 2009). Furthermore, our co-authors have detected this issue in all three of their Halo lidars at OU/NSSL. In the figure below, we show how the vertical velocity variance remains nearly constant for one range-gate, while it varies for the other ones. Therefore, we decided to remove data from the lowest three range gates in the Halo lidar.

[Figure]

**Minor comments**

*- line 50: which type of the HATPRO are you using? would be useful and consistent with the type for the lidar*

Thank you for your suggestion on adding this information. We added a reference where the interested reader can find more information. We did not include further information because this is not directly related to our dataset and manuscript.

*- line 50: is the Atmospheric Emitted Radiance Interferometer "home-made" or do you also have manyufacturer and type for it?*

Thank you for your suggestion on adding this information. We have a reference where the interested reader can find more information about the AERI. We did not include further information because this is not directly related to our dataset and manuscript.

*- Fig. 1: an additional overview map on the location on a bit larger scale would be desireable! And I also would highly prefer the x and y axes labeling in km instead of degree*

Thank you for your comment, we agree on labeling x- and y-axes with distances in km. We modified our map to show a larger portion of the terrain around Moffat School and Saguache. We also modified the labeling to km instead of degree.

*- Fig. 6: I assume you have an issue with artificially enhanced dissipation rates by the Halo lidar that is related to the strange bright band in the figure 4a*

Thank you for this comment, as we mentioned above, we discarded data within the first three range gates of the Halo lidar.

*- references: inconsistencies in abbreviating/not-abbreviating journal names*

Thank you for this comment, we fixed this issue. All of our references have their corresponding abbreviated journal names.

---

## Author Comment (AC2)

**Response to reviewer 2**

Dear Anonymous Reviewer 2,

we highly appreciate your feedback. It helped us to improve the manuscript. Below we comment on your suggestions in detail.
All reviewer comments appear in *italic text* below, while authors' responses appear in blue text. Line numbers referenced in the authors' responses refer to the revised document.

*This is likely to be a valuable addition to the data corpus of environmental turbulence measurements. I have two general recommendations, one superficial the second of more concern.*

**General comments**

*(1) Minor. The Introduction and in part the summary both refer to the field campaign being part of LAPSE-RATE which was focused on remotely piloted aircraft (RPA). The implication is that paper uses data from instruments on RPA platfroms. As far as I can see, this may be a later intention, but for this paper it is not relevant.*

Thank you for this suggestion. However, we believe including information about LAPSE-RATE is highly important since the data collection took place within this campaign and this dataset can then be used for comparisons with data from RPA platforms in the future.

*(2). Major. The paper describes and compares two lidar systems, both used to estimate turbulent dissipation. A reader coming to this paper and data set would wish to know (a) Are the instrument systems actually fit for purpose to do this, and (b) are these data useful. This is not possible to judge because there is no indication of error analysis or displays of confidence limits or other typical presentations when measurement sets (whether instrument or model output) are compared.*

We carefully considered this suggestion and included a new section dedicated to uncertainty analysis of our calculations (Section 5 in revised version of manuscript). We used the law of combination of errors to evaluate how random errors propagate through our calculations. We also compare the uncertainty in $\varepsilon$ retrievals with the corresponding values of turbulence dissipation rate for each lidar.

*Figures 2 and 3 are noteworth here: as far as I can see, figure 2 is smoothing of a noisy curve (using limted splines), whist figure 3 is fitting a Butterworth-style transfer with pre- defined cutoff (-5/3). There is no knowledge gained from these. I recommend some estimate (with error) of say the displation decay, and whether it agrees or not with Kolmagorov. Only when we have these statistical results can the quality and benefit of these data and methods be assessed by the reader.*

Thank you for your thoughtful comment. Building on your comment, we modified the method used to calculate the sample size from the energy spectrum for each 15-minute time period. We decided to do a power fit to the data at increasingly larger frequencies and then compared our results with the Kolmogorov $f^{-5/3}$ law. We then determined the transition frequency for each spectrum from the closest agreement between our power fits and Kolmogorov. Using this method, not only are we forcing agreement with Kolmogorov, but also our results agree with larger transition frequencies during nighttime and smaller transition frequencies during daytime.